# Response of Normal Tissues to Boron Neutron Capture Therapy (BNCT) with ^10^B-Borocaptate Sodium (BSH) and ^10^B-Paraboronophenylalanine (BPA)

**DOI:** 10.3390/cells10112883

**Published:** 2021-10-26

**Authors:** Hiroshi Fukuda

**Affiliations:** Department of Radiology, Faculty of Medicine, Tohoku Medical and Pharmaceutical University, Sendai 983-8536, Japan; hirofuku@tohoku-mpu.ac.jp; Tel.: +81-22-290-8781

**Keywords:** BNCT, normal tissue damage, BSH, BPA, RBE, CBE

## Abstract

Boron neutron capture therapy (BNCT) is a cancer-selective radiotherapy that utilizes the cancer targeting ^10^B-compound. Cancer cells that take up the compound are substantially damaged by the high liner energy transfer (LET) particles emitted mainly from the ^10^B(n, α^7^Li reaction. BNCT can minimize the dose to normal tissues, but it must be performed within the tolerable range of normal tissues. Therefore, it is important to evaluate the response of normal tissues to BNCT. Since BNCT yields a mixture of high and low LET radiations that make it difficult to understand the radiobiological basis of BNCT, it is important to evaluate the relative biological effectiveness (RBE) and compound biological effectiveness (CBE) factors for assessing the responses of normal tissues to BNCT. BSH and BPA are the only ^10^B-compounds that can be used for clinical BNCT. Their biological behavior and cancer targeting mechanisms are different; therefore, they affect the CBE values differently. In this review, we present the RBE and CBE values of BPA or BSH for normal tissue damage by BNCT irradiation. The skin, brain (spinal cord), mucosa, lung, and liver are included as normal tissues. The CBE values of BPA and BSH for tumor control are also discussed.

## 1. Introduction

Boron neutron capture therapy (BNCT) is a cancer-targeting radiotherapy utilizing boron−10 (^10^B), which efficiently absorbs thermal neutrons (cross section: 3830 barn) and releases high LET (230 keV/μm) <œ particle and ^7^Li particles through the ^10^B (n, α) ^7^Li reaction. As the range of the particle is only approximately 10 μm, which is close to the diameter of a cell, the thermal neutron irradiation causes significant damage limited to cells that have taken up the tumor-targeting ^10^B-compound. A clinical trial of BNCT was first applied to glioblastoma multiforme [1]. A Japanese group conducted extensive clinical trials of BNCT and obtained promising results in glioblastoma using sodium ^10^B-borocaptate (BSH) [2,3], malignant melanoma using ^10^B-p-boronophenylalanine (BPA) [4,5], and head and neck cancer using BPA [6,7]. Stimulated by the promising results from the Japanese group, clinical trials of BNCT have been conducted for glioblastoma [8,9,10,11], recurrent head and neck cancers [12,13,14], and melanoma [15,16,17]. These clinical trials have been conducted using nuclear reactors. The results of these clinical trials were summarized in a review [18]. Recently, two groups in Japan conducted clinical trials of BNCT using an accelerator for recurrent head and neck cancer. Based on the good results of this clinical trial [19], the Japanese government approved the health insurance coverage of accelerator-based BNCT for recurrent or locally advanced head and neck cancer in 2020. Rather than using a nuclear reactor, an in-house accelerator is more suitable for conducting clinical practice of BNCT in terms of many regulations regarding facility management and radiation protection. Finally, the time has come to start clinical practice of BNCT. Two BNCT centers in Japan are conducting clinical practice of accelerator-based BNCT for recurrent head and neck cancer. The clinical trial of BNCT for glioma has been completed and is awaiting government approval. We are also preparing to start clinical practice of BNCT at two other sites in Japan.

To fully understand BNCT, it is essential to understand some radiobiological considerations. In conventional radiotherapy, fractionated irradiation is used to avoid damage to normal tissues. However, BNCT is usually performed with a single dose of neutrons. Therefore, it is necessary to determine the permissible dose to normal tissues from single dose irradiation and the tumor control dose from single dose irradiation. Another difficult task is to determine the relative biological effectiveness (RBE) and compound biological effect (CBE) factors of BNCT irradiation. In neutron irradiation, high-LET particles emitted from ^14^N (n, p)^14^C and ^10^B (n, α) ^7^Li reactions are mixed with low-LET gamma rays emitted from ^1^H (n, γ) ^2^H reactions and from the reactor core and structural materials. The proportion of each radiation component varies with the depth from the body surface [20]. The quality of the neutron beam affects the depth-dose change of each radiation component. The different energy spectra of neutrons, such as pure thermal neutrons or epi-thermal neutrons, also influence the biological effect as well as the depth-dose change. However, these aspects will not be discussed in detail in this article. Furthermore, biological effect is largely determined by the microdisribuiton of ^10^B compounds in cells and tissues. For the purpose of evaluating this point, the CBE factor was introduced by Morris et al. [21]. In other words, CBE is an RBE value that takes into account the biodistribution of the boron compounds in cells and tissues, and the CBE value varies depending on the boron compounds used. It is important to estimate the dose not only at the macroscopic level but at the microscopic level. These complex conditions make it difficult to determine the CBE values for normal tissue damage as well as tumor control by BNCT. The definitions of RBE and CBE are described in Equations (1) and (2), respectively.
RBE _thermal beam_ = (X-ray ED_50_ − γ-ray dose)/thermal beam ED_50_(1)
where ED_50_ is the physical absorbed dose which results in a 50% incidence of a biological endpoint. Thermal beam includes protons from ^14^N(n, p)^14^C reaction and protons from fast neutron recoils.
CBE = X-ray ED_50_ − (thermal beam ED_50_ × RBE)/^10^B (n, α) ^7^Li ED_50_(2)

In this review, we summarize and discuss the normal tissue damage caused by BNCT irradiation with BSH or BPA. The normal tissues covered in this article are the skin, brain (spinal cord), mucosa, lung, and liver.

## 2. Boron Delivery Agents and Their Biodistribution

To date, a number of promising ^10^B compounds for BNCT have been developed [22], but only two, BSH and BPA, have been used for clinical BNCT. In this section, the differences in the biological behavior of BSH and BPA that may affect the CBE values are discussed.

### 2.1. BSH

BSH (Na_2_B_12_H_11_SH) has been used for BNCT for more than 50 years. BSH was first introduced by Soloway and Hatanaka in 1967 [23] and used for BNCT of glioblastoma multiforme (GBM) by Hatanaka in 1968. BSH does not pass through the intact blood–brain barrier (BBB) and therefore ^10^B-concentration in the normal brain is very low. In contrast, in malignant brain tumors, BSH accumulates in brain tumors due to BBB breakdown. In a European BNCT phase I study (EORTC 11961), to evaluate the biodistribution of BSH, 14 patients with gliobastoma were administered with 100 mg/kg BSH. The average normal brain-to-blood ratio was 0.2 ± 0.02 and the tumor-to-blood ratio was 0.6 ± 0.2 [24]. Brain damage was considered to be due to vascular endothelial cells damage by high-LET ^10^B (n, α)^7^Li reactions released within the vascular lumen. As the thickness of the vascular endothelial cell is at 1–2 μm level, which is smaller than the range particles, some of the radiation emitted from the ^10^B (n, α)^7^Li reaction passed through without being absorbed. Kitao [25,26] calculated the absorbed dose fraction near the interface between boron-containing and boron-free regions from the ^10^B (n, α)^7^Li reaction. The results showed that the absorbed dose fraction to the endothelial cells was 0.3–0.4 of the total doses released from the ^10^B (n, α)^7^Li reaction. This factor is very important in determining the CBE value of BSH for normal brain damage. As described in the following section, the CBE values of BSH for normal brain damage ranged from 0.3–0.5.

### 2.2. BPA

BPA is a derivative of the neutral amino acid phenylalanine, and was first synthesized by Snyder et al. in 1958 [27]. BPA having a chemical structure similar to that of tyrosine, which is the precursor amino acid of melanin polymers, was re-evaluated by Yoshino and Mishima and found to be a good melanoma-seeking ^10^B-compound in 1986 [28]. As a result of enhanced cellular amino acid transport, BPA can be taken up by any kind of cancer cell, although it remains longer in melanoma cells than in other cancer cells by interacting with the melanin polymerization process of melanoma cells [29]. Based on basic and preclinical data, Mishima et al. initiated a clinical BNCT in 1987 for malignant melanoma using BPA [4,5]. Fukuda et al. evaluated the biodistribution of BPA in patients with malignant melanoma [30]. After administration of 170–210 mg/kg BPA in the melanoma patients, ^10^B concentrations in the blood and tumors were measured. The results showed that the mean skin-to-blood ratio was 1.31 ± 0.22, and the tumor-to-blood ratio was 3.40 ± 0.83. BPA was also used for BNCT of glioblastomas [8,9,10,11]. In contrast to BSH, BPA is actively transported across the BBB and taken up by the normal brain. Tumor-to-blood and brain-to-blood ratios in human GBM patients were 1.5 to 2.4 and 0.7 to 1.0, respectively [31,32]. BPA selectively accumulates in dividing cells and not in non-dividing [33] or quiescent (Q) tumor cells [34]. Therefore, BNCT with BPA may lead to recurrence even if the initial tumor control is good. Conversely, BSH is relatively homogenously distributed in cells and tissues, although ^10^B concentration in the tumor is relatively low. To overcome the drawbacks of the two compounds, Ono et al. [35] investigated the combination of BSH and BPA in BNCT using an experimental tumor, and demonstrated improved therapeutic effects with TCD_50_ analysis. Miyatake et al. [36] performed BSH-BPA BNCT on patients with malignant glioma and achieved good tumor control.

## 3. Normal Tissue Damage due to BNCT Irradiation Using BSH or BPA

BNCT causes relatively little damage to normal tissues, but as with conventional radiotherapy, treatment must be performed within the tolerable dose limits of normal tissues. The normal tissues in question are the brain and skin in the case of glioblastoma, skin in the case of malignant melanoma, and skin, mucosa, and bone in the case of head and neck cancer. The RBE and CBE values were calculated by taking the ratio of physical dose from the thermal neutron component and the dose from ^10^B (n, α)^7^Li reactions to the gray-equivalent (Gy-Eq) dose by X-ray required to achieve the same endpoint. There is an excellent review by Coderre et al. on skin and brain damage and RBE and CBE values by BNCT [37], but we have added data on mucosa, lung, and liver, and discussed them in detail.

### 3.1. The Skin

Table 1 shows the RBE and CBE of the skin when BNCT was performed with BPA or BSH.

Hiratsuka et al. [38] evaluated skin reaction of hamsters after BPA-BNCT irradiation. The values of RBE and CBE were determined by comparing the effect of the electron beam with the physical dose by BNCT, using the moist desquamation of the skin as an end point. The RBE of the thermal neutron component without BPA was 2.2 ± 0.06. The CBE values for BPA were 2.4 ± 0.06. Only human skin data on BPA were reported by Fukuda et al. (1994) [39]. They analyzed the skin reactions of 18 patients with malignant melanoma who had received BPA-BNCT. The CBE of BPA assessed by moist desquamation was 2.5, which has been generally used as the CBE value of the human skin in BNCT clinical trials. The CBE for rat skin was 3.73 [21], which was much higher than that for hamster and human skin. This may be due to the very high X-ray dose (42 Gy-Eq) required to yield moist desquamation in the rat skin. The CBE values for BSH were 0.56 in rats [21]. The CBEs of BPA and BSH with skin necrosis as the end point were 0.73 ± 0.42 and 0.86 ± 0.08, respectively, which are almost similar. In contrast, when moist desquamation was used as the end point, there was a large difference in CBE values between two compounds (3.74 and 0.56, respectively). In dermal necrosis, vascular endothelial cells may be the target cells. Coderre et al. [42] discussed that BPA and BSH have similar micro-distribution at the dermal level, and that BSH-mediated BNCT causes less damage to the epidermis than BPA-mediated BNCT. BNCT using these two compounds causes almost the same level of damage to the dermis.

The CBE values of sodium pentaborate (Na_2_B_10_O_16_), which was used in the initial phase of BNCT for glioblastoma, were 1.87 in rabbits [40] and 2.2 in pigs [41]. These values are close to those of hamsters and humans for BPA.

### 3.2. The Central Nervous System (CNS)

Table 2 summarizes the damage to CNS (brain and spinal cord) caused by BNCT radiation with BPA or BSH.

The endpoints for brain damage were brain necrosis (BN) and magnetic resonance imaging (MRI) signal changes, while the end point for spinal cord damage was paralysis. The CBE values for BPA for rat spinal cord (paralysis) were 1.33 ± 0.16 [43] or 1.34 ± 0.13 [44]. The CBE values of BPA on the canine brain were 1.2 for brain necrosis (BN) and 1.1 for MRI change [45]. CBE values varied from 1.1 to 1.34 and were consistent between the two different tissue of interest (spinal cord and brain) and different end points. Only human data for BPA were reported by Coderre et al. [46]. They analyzed neurological symptoms of 71 patients with glioblastoma who received BPA-BNCT at the Brookhaven Medical Research Reactor (BMRR) and Massachusetts Institute of Technology Reactor (MITR). Somnolence, one of the neurological symptoms defined as the state of feeling drowsy, was used as an end point. Since the X-ray dose at which somnolence occurred in humans was unknown, the RBE and CBE values in animals reported by the authors [36,43,44] were used for the calculation of Gy-Eq doses. The calculated doses with 50% incidence of somnolence (ED_50_) were 6.2 ± 1.0 Gy-Eq for average brain dose and 14.1 ± 1.8 Gy-Eq for peak brain dose. Unfortunately, MRI findings of the patients were not included in this report.

The CBE value of BSH for paralysis in rats was 0.46 ± 0.05 [43] or 0.53 ± 0.03 [47]. Damage to the canine brain was assessed using brain necrosis (BN) or MRI change as end points. The CBE values of BSH for the canine brain were 0.37 for brain necrosis and 0.27 for MRI change [45] when BMRR was used. In contrast, the CBE values for brain necrosis and MRI change were 0.55 and 0.49, respectively, and were different from the values obtained at BMRR, when the High Flux Reactor in Petten (HFR) was used. Gabel et al. evaluated the CBE values of BHS for canine brain damage using neurological symptoms (ataxia, disorientation, and inability to walk) and MRI change as end points. The results showed that the CBE values of BSH in the canine brain were 0.37 and 0.66, respectively [48].

### 3.3. The Mucosa (Tongue)

Table 3 shows the RBE and CBE of BPA and BSH on the mucosa obtained using the rat tongue model after BNCT irradiation.

Coderre et al. evaluated the CBE of BPA on the rat tongue with 50% area ulceration as the endpoint and found the CBE value to be 4.9 [49]. Using the same model, Morris et al. [50] also reported a CBE value of 4.87 ± 0.16 for BPA and 0.29 ± 0.02 for BSH. The CBE values of BPA for mucosa were much higher than those for the skin. This may be explained by the higher radiosensitivity of mucosa to conventional radiotherapy compared to skin.

### 3.4. The Lung

Table 4 shows the CBE of BPA for lung injury due to BNCT irradiation reported by Kiger et al. [51,52].

An increase in respiratory rate of 20% or more after irradiation was used as an index for evaluation. Early and late injuries were defined as injuries occurring before and after 110 days after irradiation, respectively. The radiation doses resulting in early and late injury by X-ray were 11.4 ± 0.4 Gy and 11.6 ± 0.4 Gy, respectively, which were almost the same. The CBE values for early and late injury were 1.4 ± 0.3 and 2.3 ± 0.3, respectively. The reason for the higher CBE of late failures despite the doses producing early and late failures by X-rays being almost the same is unknown, but it may be that BNCT is more damaging to the target cells of late injury. There are no reports on the CBE of BSH.

### 3.5. Liver

The only report on hepatocellular damage after BNCT irradiation was that by Suzuki et al. [53] and the results were shown in Table 5. After administration of BSH or BPA, mice were irradiated with thermal neutrons. Immediately after irradiation, partial hepatectomy was performed to stimulate hepatocyte division. Five days after the procedure, the liver was removed following through perfusion of the inferior vena cava with 0.25 % trypsin, minced, and filtered with a mesh. Dispersed cells were prepared for micronucleus assay. A survival curve was generated with the percentage of cells without micronucleus on the vertical axis and the irradiation dose on the horizontal axis. The CBE values were calculated by comparing the slope (D_0_) of the curve with results obtained with X-rays. The CBE values were 4.25 for BPA and 0.94 for BSH [53].

## 4. Tumor Control by BNCT

The values of RBE and CBE for tumor control were first reported by Hiratsuka et al. [54]. Hamsters bearing Green’s melanoma were treated with BPA and irradiated with thermal neutrons. Tumor volume was measured daily, and growth delay time was used as the end point. The results were compared with those obtained using an electron beam. The RBE values of the neutron beam with BPA and that of the ^14^N(n, p)^14^C reactions were 2.22 and 3.0, respectively, while the CBE value of ^10^B(n, α)^7^Li was 2.5. Coderre et al. [55] evaluated the effect of BNCT on Harding-Passey melanoma in vivo using growth delay time as an index, and found that RBE value of BNCT irradiation was approximately 2.0 compared to the effect by 100 keV X-ray. They did not report the CBE value of ^10^B (n, α)^7^Li reaction. In a later study, Coderre et al. [56] reported the in vivo effect of BNCT on B-16 melanoma using a morbidity index obtained from survival curves. For the calculation of Gy-Eq dose, 2.3 and 2.0 were used for the RBE values of ^10^B(n, α)^7^Li and ^14^N (n, p)^14^C reactions, respectively, based on the literature. Coderre et al. [57] evaluated the effect of BNCT on Fisher 344 rats with 9 L gliosarcoma in the brain. After irradiating rats with thermal neutrons, brain tumors were removed, minced, and trypsinized to prepare single-cell suspension. These cells were plated in a culture flask and a colony formation assay was performed. Comparing these with results obtained using 250 keV X-rays, the CBE value of ^10^B(n, α)^7^Li reaction was determined to be 3.8 at survival fraction level of 0.01. Suzuki et al. [53] evaluated the effect of BNCT on a SCC VII tumor implanted in the liver of C3H/He mice. After intraperitoneal administration of 75 mg/kg BPA, the mouse tumor was irradiated with thermal neutrons in vivo and then assayed in vitro using the same method described above. The CBE value of ^10^B(n, α)^7^Li reaction for BPA was 4.22 by the D_0_ ratio of the survival curve. The CBE value of BSH was 2.29. These results were summarized in Table 6.

## 5. Discussion

The CBE values summarized in this article were calculated assuming that the RBE value of a thermal (epithermal) neutron beam is constant regardless of radiation doses. However, the RBE of high-LET radiation increases with a decreasing dose. The biological effect of a thermal neutron beam is mainly due to the high-LET ^14^N (n, p)^14^C reaction and, therefore, the dose-dependent changes in the RBE of the thermal neutron beam should be taken into account. Morris et al. [58] published an important article on the re-evaluation of CBE values. They reported that the RBEs of the thermal neutron beam at the Brookhaven Medical Research Reactor (BMRR) varied from 2.57 to 1.40 depending on the fraction of thermal beam component of total doses, including the dose from a ^10^B (n, α)^7^Li reaction. Previously, a fixed value of 1.4 ± 0.04 has been used as the RBE value of a thermal neutron beam. This value was obtained by thermal neutron irradiation without the ^10^B-compound. Using these variable RBE values, they re-calculated the CBE values of the ^10^B (n, α) ^7^Li reaction for the BPA of spinal cord damage. The original CBE value of 1.33 ± 0.16 (Table 2 in this article) was adjusted to 0.88 ± 0.14. The RBE-adjusted CBE values decreased from 0.88 to 0.48 depending on the ^10^B concentration in the blood’s increase. However, from a clinical perspective, the use of RBE-adjusted CBE for dose calculation is not recommended because there is still uncertainty over accurate determination of RBEs of thermal neutron beams.

Ono [59] recalculated the RBE and CBE of BHS for CNS damage using the data from Morris et al. [60]. The recalculated RBE of the thermal neutron beam varied from 1.62 to 2.50 and RBE-adjusted CBE values of BHS for CNS damage were 0.36 ± 0.02 while the original CBE value was 0.52 ± 0.02. In his article, Ono also analyzed the structure of the CBE factor. To describe the CBE structure, the following terms were introduced: the vascular CBE (v-CBE), intraluminal CBE (il-CBE), extraluminal CBE (el-CBE), and non-vascular CBE (nv-CBE) factors, and geometric biological factor (GBF). Using the results by Morris et al. [60], Ono [59] analyzed the relationship between the CBE factor of BPA to the CNS damage and the normal tissue/blood ratio and found a good linear correlation between them. Base on this linear regression line, the CBE value of BPA for CNS damage can be calculated by the following formula.
CBE factor = 0.32 + N/B × 1.65(3)
where N/B is normal tissue/blood ^10^B concentration ratio. This formula indicates that the CBE value depends on the N/B ratio and not on absolute ^10^B-concentraion either in the blood or in the normal tissue. Ono also proposed a formula for the estimation of late damage to the skin and lung.
CBE factor = 0.32 + N/B × 1.80(4)

In this article, different responses of various normal tissues are summarized. RBE values of the thermal neutron beam (RBE_beam_) for different normal tissues are shown in Table 1, Table 2, Table 3, Table 4 and Table 5. The biological effect of the beam is mainly caused by ^14^N(n, p)^14^C reactions. The values were relatively high in the skin (2.5, 3.5) and low in the CNS (1.4, 2.1), in the liver (1.37), and in the lung (1.2). The variations may be due to different radio-sensitivities of each tissue, as is the case in other types of radiation. CBE values ranged from 1.3 to 4.9 by BPA and 0.3 to 0.9 by BSH. The large variation may be caused by different micro-distribution of the boron compound utilized as well as the different sensitivities of various normal tissues.

Accelerator-based BNCT utilizes epithermal neutrons which penetrate into deeper parts of the body compared to thermal neutrons. Due to the differences in the energy spectrum and depth-dose distribution, the value of the RBE_beam_ may be different from that of thermal neutrons. However, this point is not discussed in detail in this article.

## 6. Conclusions

In addition to the many difficulties described in the introduction in understanding the radiobiological basis of BNCT, another factor affecting CBE values, the variable RBE values of the thermal neutron beam component, is described in the discussion. Although it will be very difficult for clinical oncologists to fully understand these complex radiobiological data, I believe that careful observation and recording of not only tumor response but also normal tissue responses in human patients, and analyzing them based on radiobiological principles, will lead to further development and widespread application of clinical BNCT.

In order to promote BNCT, it is necessary to educate and train young radiation oncologists who understand the complex radiobiological aspects of BNCT. In addition, it is necessary to improve the educational system for training BNCT specialists and the qualification system for specialists. The Japanese Society for Neutron Capture Therapy (ISNCT) is currently beginning to experiment with such a system.

## Figures and Tables

**Table 1 cells-10-02883-t001:** RBE and CBE for the skin after BNCT radiation using BPA or BSH.

Compound	Exp. System	End Point	Dose ^1^ (Gy-Eq)	RBE_beam_ ^2^	RBE _(n, p)_ ^3^	CBE	Administered Dose (mg/kg)	Reference
BPA	Hamster	No more than MD	24.0 (electron)	2.2 ± 0.06	2.9 ± 0.04	2.4 ± 0.06	10, 20, 40,80, ip	Hiratsuka, 1991 [38]
BPA	Human	MD	18.0	n.a	2.5	2.5	170–210, iv	Fukuda, 1994 [39]
BPA	Fischer rat	MD DN	42.15 ± 2.24 56.6 ± 2.9	3.5 ± 0.23 (3.5)	n.a	3.74 ± 0.70 0.86 ± 0.08	1500, oral	Morris, 1994 [21]
BSH	Fischer rat	MD DN	42.15 ± 2.24 56.6 ± 2.9	3.5 ± 0.23 (3.5)	n.a	0.56 ± 0.06 0.73 ± 0.42	100, iv	Morris, 1994 [21]
Na_2_B_10_O_16_	Rabbit	Grade 3 (MD)	23.3	2.5	3.9	1.87	35, iv	Yamamoto, 1961 [40]
Na_2_B_10_O_16_	Swine	MD	22.7	1.5 ± 0.3	2.7	2.2	35, iv	Archambeau, 1971 [41]

^1^: Required X-ray equivalent dose (Gy-Eq) for inducing the end point; ^2^: RBE of thermal beam com. ponents; ^3^: RBE of ^14^N (n, p)^14^C reaction; MD: moist desquamation; DN: dermal necrosis; n.a: not available; ip: intraperitoneal administration; iv: intravenous administration; Na_2_B_10_O_16_: sodium pentaborate.

**Table 2 cells-10-02883-t002:** RBE and CBE for the central nervous system (CNS) after BNCT radiation using BPA or BSH.

Compound	Exp. System	End Point ED_50_	Dose ^1^ (Gy-Eq)	RBE_beam_ ^2^	RBE _(n, p)_ ^3^	CBE	Administered Dose (mg/kg)	Reference
BPA	Spinal cord, Fischer rat	Paralysis	19.0 ± 0.2	1.4 ± 0.04	n.a	1.33 ± 0.16	1500, oral	Morris, 1994 [43]
BPA	Spinal cord, Fischer rat	Paralysis	19.0 ± 0.2	1.4 ± 0.04	n.a	1.34 ± 0.13	1500, ip	Coderre, 2000 [44]
BPA	Brain, retriever dog	BN MRI change	14.9 12.8	n.a	3.3^3^	1.1 1.1	950, oral	Gavin, 1997 [45]
BPA	Brain, human received BNCT	Somnolence	6.2 ± 1.0 (av) 14.1 ± 1.8(p)	-	(3.2) ^4^	(1.3) ^4^	250, 290, 330, iv	Coderre, 2004 [46]
BSH	Spinal cord, Fischer rat	Paralysis	19.0 ± 0.2	1.4 ± 0.04	n.a	0.46 ± 0.05	100, iv	Morris, 1994 [43]
BSH	Spinal cord, Fischer rat	Paralysis	19.0 ± 0.2	2.13 ± 0.06	n.a	0.53 ± 0.03	80, iv	Morris, 1997 [47]
BSH	Brain, retriever dog	BN MRI change	14.9 12.8	n.a (BMRR)	3.3^3^	0.37 0.27	50–100, iv	Gavin, 1997 [45]
BSH	Brain, retriever dog	BN MRI change	14.9 12.8	n.a (HFR)	3.3^3^	0.55 0.49	25, 50, iv	Gavin, 1997 [45]
BSH	Brain, beagle dog	N sympt. MRI change	14.9 12.8	n.a	3.93 ± 0.43 ^3^ 2.33 ± 0.14	0.37 ± 0.06 0.66 ± 0.04	not given	Gabel, 1998 [48]

^1, 2, 3^: same as in Table 1; ^4^: data from literature; ED_50_: 50% effective dose; BN: brain necrosis; av: average brain dose; p: peak brain dose; BMRR: Brookhaven Medical Research Reactor; HFR: High Flux Reactor in Petten; N sympt: neurological symptoms (ataxia, disorientation, and inability to walk).

**Table 3 cells-10-02883-t003:** RBE and CBE for the mucosa after BNCT radiation using BPA or BSH.

Compound	Exp. System	End Point	Dose ^1^ (Gy-Eq)	RBE_beam_ ^2^	RBE _(n, p)_ ^3^	CBE	Administered Dose (mg/kg)	Reference
BPA	Fischer rat tongue	Ulceration (50% area)	13.4 ± 0.2	n.a	3.2	4.9	1500, ip	Coderre, 1999 [49]
BPA	Fischer rat tongue	Ulceration (50% area)	13.4 ± 0.2	n.a	(3.20 ± 0.1) ^4^	4.87 ± 0.16	700, ip	Morris, 2000 [50]
BSH	Fischer rat tongue	Ulceration (50% area)	13.4 ± 0.2	n.a	(3.20 ± 0.1) ^4^	0.29 ± 0.02	39/50, ip	Morris, 2000 [50]

^1, 2, 3, 4^: the same as in Table 1; n.a: not available; ip: intraperitoneal administration.

**Table 4 cells-10-02883-t004:** RBE and CBE for the lung after BNCT radiation using BPA or BSH.

Compound	Exp. System	End Point	Dose ^1^ (Gy-Eq)	RBE_beam_ ^2^	RBE _(n, p)_ ^3^	CBE	Administered Dose (mg/kg)	Reference
BPA	Fischer rat	BR increase ≥ 20%	11.6 ± 0.1	1.2	2.2	1.5	700, ip	Kiger, 2004 [51]
BPA	Fischer rat	BR increase ≥ 20%	E: 11.4 ± 0.4 L: 11.6 ± 0.4	E: 1.23 ± 0.09 L: 1.21 ± 0.08	E: 3.0 ± 0.7 L: 3.1 ± 1.2	E: 1.4 ± 0.3 L: 2.3 ± 0.3	900, ip	Kiger, 2008 [52]

^1, 2^^, 3^: the same as in Table 1; BR: breathing rate; E: early response occurring < 110 days; L: late response occurring ≥110 days.

**Table 5 cells-10-02883-t005:** RBE and CBE for the liver after BNCT radiation using BPA or BSH.

Compound	Exp. System	End Point	RBE_beam_ ^2^	RBE _(n, p)_ ^3^	CBE	Administered Dose (mg/kg)	Reference
BPA	C3H/He mice	Micronucleus assay, D_0_	1.37	n.a	4.25	1500, oral	Suzuki, 2000 [53]
BSH	C3H/He mice	Micronucleus assay, D_0_	1.37	n.a	0.94	75, iv	Suzuki, 2000 [53]

^2, 3^: the same as in Table 1.

**Table 6 cells-10-02883-t006:** RBE and CBE for tumor control by BNCT radiation using BPA.

Compound	Tumor Line	End Point	Dose ^1^ (Gy-Eq)	RBE_beam_ ^2^	RBE _(n, p)_ ^3^	CBE	Administered Dose (mg/kg)	Reference
BPA	Green’s melanoma	Growth delay time		2.22	3.0	2.5	10, 20, 40, ip	Hiratsuka, 1989 [54]
BPA	Harding- Passey	Growth delay time	TCD_50_ 29 ± 3	2.0	n.a	n.a	300, ip 1500, oral	Coderre, 1988 [55]
BPA	B-16	Morbidity index	TCD_50_ 29	2.0	(2.0)^4^	(2.3) ^4^	1500, oral	Coderre, 1991 [56]
BPA	9L-gliosarcoma	CFA ratio at SF = 0.01	-	2.3	3.2	3.8	1500, oral	Coderre, 1993 [58]
BSH	9L-gliosarcoma	CFA ratio at SF = 0.01	-	-	-	1.2	50, iv	Coderre, 1993 [58]
BPA	SCC VII	CFA D_0_ ratio	-	2.79	-	5.64	1500, oral	Suzuki, 2000 [53]
BSH	SCC VII	CFA D_0_ ratio	-			2.29	75, iv	Suzuki, 2000 [53]

^1, 2, 3, 4^: the same as in Table 1; Green’s melanoma: hamster melanoma; Harding-Passey: murine (Balb/c) melanoma; B-16: murine (C57Black) melanoma;9L-gliosaracome: rat (Fishier) gliosarcoma; SCC VII: murine (C3H/He) squamous cell carcinoma; Ratio at CFA: After irradiation of tumors in vivo, cell suspension was made and then assayed by colony formation (CFA); SF: survival fraction; D_0_: slope of the survival curve.

## Data Availability

Not applicable.

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
