# Peer review of "Response of Normal Tissues to Boron Neutron Capture Therapy (BNCT) with 10B-Borocaptate Sodium (BSH) and 10B-Paraboronophenylalanine (BPA)"

_cells, 2021, doi:10.3390/cells10112883_

Round 1

Reviewer 1 Report

In this MS, the author provides a comprehensive review of the effects of BNCT with BPA or BSH on normal tissues based on previous literatures.

The findings summarized in this MS are important for a proper understanding of future BNCT studies.

In particular, it is important for future clinicians working on BNCT to understand the effects of BNCT on normal tissues, as well as on cancer tissues.

In addition, knowledge of the effects of BNCT on normal tissues should always be taken into account in the development of boron agents for BNCT.

For these reasons, I believe that the information in this MS is very useful.

Author Response

Thank you for your nice comments on my manuscript. I revised it according to the reviewer 2 and 3. 

Reviewer 2 Report

With the development of modern technology, BNCT is becoming one of the promising radiotherapies for cancer treatment. In this manuscript, Prof. Fukuda gives a thorough review on the response of normal tissues to BNCT. Overall, the author summarizes the RBE/CBE of various tissues, which serve valuable reference for both clinical applications and basic science.

I recommend the publication with minor revisions (see below).

Line48-49. “Another difficult task is to determine the relative biological effectiveness (RBE) and compound biological effect(CBE) factors of BNCT.”

The distribution of 10B is a critical factor for BNCT, and the definition of CBE seem vary in different study; in this manuscript, the response of normal tissues to BNCT are depicted using either RBE or CBE. It would be appreciated if the author can elaborate the definition of RBE and CBE in more detail.

Line 52-53. “The proportion of each radiation…varies with the depth from the body surface.”

Please add reference here, though the concept seems to be broadly accepted.

Especially, the quality of thermal neutron serves an important role in BNCT.

Line 80. Some of the “adiation”

Line 277. “CBE value of BPA can be calculated …”

In this manuscript, different responses of various normal tissues are summarized. Is any conclusion regarding the tissue-dependence? If so, few more discussions regarding mechanisms would be appreciated.

Line 284. “the variable RBE values of thermal neutron beam component is described in the discussion”

If possible, could author highlight the contribution of neutron beam component in more detail?

The quality of neutron is one of critical factors in BNCT, especially with the development of accelerator-based BNCT, the beam component is worthy more attention for optimizing BNCT.

Author Response

Reply to the reviewer 2

Thank you for the helpful advice to improve the manuscript. Point-by-point replies were described below.

Line 48-49.

The distribution of 10B is a critical factor for BNCT, and the definition of CBE seem vary in different study; in this manuscript, the response of normal tissues to BNCT are depicted using either RBE or CBE. It would be appreciated if the author can elaborate the definition of RBE and CBE in more detail.

Reply

Definitions of RBE and CBE are described by equation (1) and (2) in the last part of introduction (Line 71-75 in the revised manuscript).

The definition of RBE and CBE are described in equation 1 and 2, respectively.

RBE thermal beam = (X-ray ED50  - g-ray dose) / thermal beam ED50

(1)

Where ED50 is the physical absorbed dose which results in a 50% incidence of a biological endpoint. Thermal beam includes protons from 14N (n, p)14C reaction and protons from fast neutron recoils.

CBE = X-ray ED50-(thermal beam ED50 × RBE) / 10B (n, a) 7Li ED50

(2)

Line 52-53  

“The proportion of each radiation…varies with the depth from the body surface.” Please add reference here, though the concept seems to be broadly accepted. Especially, the quality of thermal neutron serves an important role in BNCT.

Reply

  • Reference

I have added the following article as a reference [20]. Figure 2 in this article shows depth-dependent change of fraction of each radiation component.

  1. Gabel, D., Present status and perspective of boron neutron capture therapy. Radiother. and Oncol. 1994, 30, 199-205, doi.org/10.1016/0167-8140(94)90458-8.
  • Quality of thermal neutron

The following sentence was added just after reference [20].(Line 60-63)

Beam quality of the neutron beam affect to the depth-dose change of each radiation component. Different energy spectrum of neutrons, such as pure thermal neutrons or epi-thermal neutron also affect the biological effect as well as the depth-dose change. However, these aspects will not be discussed in detail in this article.

Line 80. Some of the “adiation” ->radiation

Line 277 “CBE value of BPA can be calculated …”

In this manuscript, different responses of various normal tissues are summarized. Is any conclusion regarding the tissue-dependence? If so, few more discussions regarding mechanisms would be appreciated.

Reply

The following sentences were added (Line 298- to 303).

In this article, different responses of various normal tissues are summarized. RBE values of thermal or epithermal neutron beam (RBEbeam) for different normal tissues were shown in Table 1 to 5. The biological effect of the beam is mainly caused by 14N(n, p)14C reactions. The values were relatively high in the skin (2.5, 3.5) and low in the CNS (1.4, 2.1), in the liver (1.37) and in the lung (1.2). The variations may be due to different radio-sensitivities of each tissue as is the case in other types of radiations.

On the other hand, CBE values ranged from 1.3 to 4.9 by BPA and 0.3 to 0.9 by BSH. The large variation may be caused by different micro distribution of the boron compound utilized as well as the different sensitivities of various normal tissues.

Line 284. “the variable RBE values of thermal neutron beam component is described in the discussion”

Reply

I discussed it in the introduction as described above. I also added the following sentences (Line 308-311).

Accelerator-based BNCT utilize epithermal neutrons which penetrate into deeper part of the body compared to thermal neutrons. Due to the differences in the energy spectrum and depth-dose distribution, the value of RBEbeam may be different from that of thermal neutrons. However, this point is not discussed in detail in this article.

Reviewer 3 Report

The author gives a comprehensive review of boron neutron capture therapy with 10B-borocaptate sodium (BSH) and 10B-paraboronophenylalanine (BPA) regarding their biodistribution, tumor suppression and off-target effects of different tissues. Meanwhile, the author provides the detailed review of the different therapeutic value and challenges of the therapy. However, there are some minor recommendations needed resolved before the final acceptance of the manuscript. 

  1. The readers would like to know the current progress of the capture therapy and the possible solutions to the certain issues mentioned in the review paper. For example, what would be possible solutions to the issue that "clinical oncologists may not understand the radiobiological knowledge".
  2. There are many typos and confusing English writing. There are some listed below. 
    1. "... afffected..." should be "... affected..."
    2. "... ealuating..." needs to be changed to "... evaluating ..."
    3. "... dut..." change to "... due..."
    4. "... adiation..." should be "radiation".
    5. "... syhthesized ..." change to "... synthesized..."
    6. "... , and bone..." can be "... and bone..."

Author Response

Reply to the reviewer 3

              Thank you for the helpful advice to improve the manuscript. Point-by-point replies were described below.

Comment 1.

  1. The readers would like to know the current progress of the capture therapy and the possible solutions to the certain issues mentioned in the review paper. For example, what would be possible solutions to the issue that "clinical oncologists may not understand the radiobiological knowledge".

Reply

  • Current progress of the capture therapy

The following sentences which describe the current progress of BNCT in Japan was described in the introduction section (Line 43-49).

“Rather than using a nuclear reactor, an inhouse accelerator is more suitable for conducting clinical practice of BNCT in terms of regulations regarding facility management and radiation protection. Finally, the time has come to start clinical practice of BNCT. Two BNCT centers in Japan are conducting clinical practice of accelerator-based BNCT for recurrent head and neck cancer. The clinical trial of BNCT for glioma has been completed and is awaiting government approval. We are also preparing to start clinical practice of BNCT at two other sites in Japan.”

  • Possible solutions to the issue that "clinical oncologists may not understand the radiobiological knowledge".

The following sentences were added (line 303-307).

In order to promote BNCT, it is necessary to educate and train young radiation oncologists who understand the complex radiobiological aspects of BNCT. In addition, it is necessary to improve the educational system for training BNCT specialists and the qualification system for specialists. The Japanese Society for Neutron Capture Therapy (ISNCT) is currently beginning to experiment with such a system.

  1. Type errors 

I am sorry for many careless mistypes. The mistypes were corrected shown by red characters.

(x) Extensive editing of English language and style required

   Before submission, I had my manuscript checked and revised by a professional English editor. I paid for it. If it is possible to improve English writing by the editorial office, I can pay for it.
